# Consumer’s Willingness to Pay a Premium for Organic Fruits in China: A Double-Hurdle Analysis

**DOI:** 10.3390/ijerph16010126

**Published:** 2019-01-05

**Authors:** Lijia Wang, Jianhua Wang, Xuexi Huo

**Affiliations:** 1State Key Laboratory of Grassland Agro-Ecosystems, Key Laboratory of Grassland Livestock Industry Innovation, Ministry of Agriculture and Rural Affairs, Lanzhou University, Lanzhou 730020, China; wanglijia@lzu.edu.cn; 2College of Pastoral Agriculture Science and Technology, Lanzhou University, Lanzhou 730020, China; 3School of Business, Jiangnan University, Wuxi 214122, China; 4College of Management and Economics, Northwest A&F University, Yangling, Xi’an 712100, China; xuexihuo@nwsuaf.edu.cn

**Keywords:** organic fruits, China, double-hurdle model, premium, willingness to pay

## Abstract

The aim of the paper was to assess how consumers evaluate organic labeled fruits and to what extent they are willing to pay a premium for fresh fruits with organic labels. A double-hurdle model is applied to data obtained by interviewing 407 fresh fruit consumers in nine Chinese cities. Willingness-to-pay a premium was modeled as a function of a series of demographic, socio-economic variables, plus fruit attributes, perceptions of fruit safety, and risk attitudes. Results indicate that the most important factors influencing willingness to pay a premium involved positive attitudes toward organic label, attention to fruit safety, the perception of importance of fruit attributes. Moreover, the more income consumers earn, the more likely they would be willing to pay a premium for organic fresh fruits. The recorded consumer interest in safety and quality of fresh fruits reveals that a promising market for organic fruits could be developed by an adequate knowledge on organic label and an effective market monitoring system.

## 1. Introduction

An outbreak of *Escherichia coli* O157:H7 in 2006 prompted the U.S. Food and Drug Administration (FDA) to issue warnings about the safety of fresh bagged spinach and recommend consumers not to eat it [1]. In 1998, the US FDA developed guidelines to reduce microbial food hazards, particularly in the context of fresh fruits and vegetables [2]. In Europe, the “Euro-leaf” certification for organic food was launched in 1 July 2010. The use of the logo is mandatory for pre-packaged foods [3]. Organic fruits and vegetables which are free from chemicals have rapidly emerged as an important food industry in the developed countries [4]. In China, a large number of consumers panicked about all kinds of dairy products, particularly milk powder, after the 2008 melamine milk crisis. During the decade from January 2001 to January 2010, up to 1460 food quality safety incidents occurred [5]. The food scandals made food consumption in China very risky and consumers began to search for alternative safe food consumption options which ultimately led consumers toward organic food consumption as well [6]. 

The Chinese government has approved a series of tougher food safety laws and regulations to keep the food supply safe [7]. In 2010 and 2011, the Special Operation against Quality Safety Problems of Agricultural Products led by the Ministry of Agriculture and the Special Operation Combating Illegal Food Additives carried out by the State Administration of Food and Drug Safety has achieved positive results. Meanwhile, organic food, green food, non-harmful products and good agricultural practices (GAP) have been successively put into use to confront the increasing severity of food safety and environmental concerns [8]. Organic products have developed quickly in China. In 2004, General Administration of Quality Supervision, Inspection, and Quarantine of PRC (AQSIQ) promulgated “Management measures of organic products certification”, and established national standards of organic products (GB/T 19630.1-19630.4-2005) in 2005. The Organic Products Regulations define specific requirements for organic products to be labeled as organic or that bear the organic agricultural product legend (logo) (Figure 1). 

With the passage of the National Organic Program (NOP) in 2002 creating uniform USDA standards, the organic food industry has increased tremendously in terms of size and product scope [9] According to Research Institute of Organic Agriculture (FiBL), by 2014 the land area of organic agriculture in China had reached 1.9 million hectares. This is just behind Australia (22.7 million hectares), Argentina (3.1 million hectares) and the United States of America (2.0 million hectares). In 2016, the organic culture area reached 2.3 million hectares, compared to 1.6 million hectares in 2015. Therefore, China occupies the third place in the world regarding the area dedicated to organic agriculture [10]. Meanwhile, the sales volume of organic products in domestic China has been continuously increasing [11]. By the end of 2015, the sales volume of organic products in domestic China had reached 129.9 billion RMB, 10.95 thousand production enterprises received certification which was officially approved by National Standard GB/T19630 Organic Products [12].

Although the organic market in China is not fully developed, the consumption of organic food has started to grow in large cities [13]. With growing numbers of consumers concerned about health, organic fruits which contribute many nutritional benefits to human health and to overcome disease hazards [14], have started to become popular among Chinese consumers. However, consumption of organic fresh fruits has not risen to the same degree, due to the extremely higher price [11]. The sales volume of organic fruits and nuts reached 8.6 billion RMB, only accounts for 6.6% of the total organic products in domestic China in 2015 [12]. Yin et al. found that 79% of Chinese consumers perceived organic food as expensive, and only 16% judged the price as reasonable [8]. 

Results of the study should help organic fruits producers and retailers to adjust marketing strategies according to market demand. Understanding consumers’ perception and attitude toward organic fruits would provide insight for policymakers in formulating regulation to restore consumer confidence. Overall, taking into consideration of the rising interest of consumers for healthy quality fruits, the article aims to understand consumers’ WTP a premium price for organic fresh fruits and elicit factors affecting their payment behavior.

The paper is structured into five sections. The next section reviews the literature on consumer’s WTP for organic food and various research approaches. Section 3 covers the econometric model. This is followed by a description of survey design, sampling procedure. The descriptive statistical results are illustrated in Section 4. The empirical results including WTP a premium choice strategy and its impact on the premium are reported in Section 5. Discussions are finally presented along with the key recommendations for action.

## 2. Literature Review

The increase in the organic food market size has led to a number of studies on understanding consumers concerns, behavior and their readiness to pay for organic food which directly influences the organic market. Price premium, the excess prices paid over and above the “fair” price that is justified by the “true” value of the product [15], may be indicators of consumer’s demand for the products [16]. Thus, willingness-to-pay (WTP) for organic products can be a good predictor of organic food demand [17]. 

Historically, issues concerning consumers’ WTP price premium for organic food have been extensively discussed. Studies had shown that age, income, education, regional difference, attitude towards certified traceability system were significant factors affecting consumer WTP a price premium for certified food [18,19]. Nilsson et al. found that concerns about food safety had significantly affected consumers’ WTP for price premiums [20]. Schott and Bernard concluded that farm size affected consumer WTP for milk products [21]. In a survey of cities in China, Zheng et al. found that consumers were willing to pay a premium for organic attributes, taking soymilk as an example [13]. A more recent survey study disclosed that Indian consumers were willing to pay relatively more for environmentally certified production practices than their Chinese or UK counterparts [22]. Van Loo et al. pointed out that consumers were willing to pay a premium of 1.193 $/lb (34.8%) for the general organic label and 3.545 $/lb (103.5%) for the USDA organic label [23]. Little research has been dedicated to analyzing Chinese consumer preferences for organic fresh produce, particularly on their willingness to pay a premium for certified fresh fruits. From a pioneering study in Japan, He et al. revealed that Japanese consumers were willing to pay 64,300 yen per year for food with higher safety guarantees imported from China on average [24]. Huang and Lee reported that Chinese respondents were willing to pay $22.0 extra per year to buy organic certified agricultural standards milk [25]. 

A variety of methodological approaches are applied to study consumers’ WTP for premium organic food. A number of studies have designed choice experiments to elicit WTP values from respondents [26,27,28]. Choice experiments allow respondents to choose among products instead of rating or ranking them, which likely simulates real purchasing situations more closely [29]. But other scholars argue that choice experiments provide several hypothetical purchasing scenarios which may lead to hypothetical bias [30]. Therefore, a combination of sensory evaluation and experimental auctions was employed to analyze consumers’ WTP [31,32,33]. Besides, other methods such as binary probit analysis [34], interval regression method [18] were applied as well.

To date, the use of the double hurdle model to identify important consumer characteristics that are associated with WTP for organic food has increased. Katare et al. used the double-hurdle model to evaluate the impact of information on consumers’ WTP for nano-packaged food products [35]. Shi et al. employed a double hurdle count data model to explore the different consumer structures in the fresh and frozen blueberry markets [36]. Studies carried out in the African food market showed that consumers were willing to pay a premium for food-safety [1]. However, this approach has seen little use to study Chinese consumers’ WTP for organic fresh fruits. 

## 3. Methods: Double-Hurdle Model

In the study, a substantial number of participants prefer not to pay a premium for organic fruits and submit a zero answer implying a zero WTP. Assuming that for each individual the decision whether or not to pay a premium price for organic fresh fruits and the decision about the amount paid to organic fruits are made independently, the double-hurdle model introduced by Cragg is applied to identify the factors that influence respondent’s WTP [37]. Because a hurdle model is a modified count model in which the two processes generating the zeros and the positives are not constrained to be the same [38]. 

Consumers in our research make two decisions with respect to the WTP: whether willing to pay a premium price for organic fresh fruits, and how much a positive amount is paid by consumers for the organic fruit. Let *C_j_* be thejth *j*-th consumer’s premium for the organic fruit, then the probability of the consumer choosing not to pay a premium price (*C_j_* = 0) can be written as:(1)$$\mathrm{Prob}(C_{j}=0)=\Phi (-\alpha_{1}\prime X_{j})$$
where Φ is the standard normal density function, *X_j_* represents a vector for consumer *j*th socio-economic demographics, attitude towards fruit safety and perception of the importance degree of various fruit attributes, *α*_1_ is a vector of coefficients. 

The second hurdle determines the effect of independent variables on *C_j_* given that *C_j_* > 0. It is with respect to how much to pay for organic fresh fruits given that the WTP consumer has decided to buy and pay. The distribution of *C_j_* conditional on being positive is truncated at zero with mean *α*_2_*X_j_* and variance *σ*^2^. The form of the second hurdle function is then:(2)$$L(C_{j}|C_{j}>0)=\frac{(1/\sigma )\Phi \left[(C_{j}-\alpha_{2}\prime X_{j})/\sigma \right]}{\Phi (\alpha_{2}\prime X_{j}/\sigma )}$$
where *α*_2_ denotes a vector of coefficients. The hypothesis that the two stages are independent decisions is tested by a likelihood ratio statistic. 

The two models can be estimated simultaneously when the error terms in the two models are assumed to be correlated, or separately when they are assumed to be independent [39]. Several research has used the same set of independent variables in the two models (hurdles), especially when data are limited [40]. In this study, the same set of independent variables in the binomial probability model and truncated-at-zero count model are employed while the two models are estimated separately largely due to data limitation. In order to interpret the percentage changes in the probability that a consumer is willing to pay a premium for organic fresh fruits due to a unit change of the explanatory variable of interest given others being constants, marginal effects are calculated in the binary Probit model [41]. With regard to dummy explanatory variables, the marginal effect evaluates the probability changes caused by a discrete change from 0 to 1 [42].

## 4. Sample Selection and Data

### 4.1. Survey Design

Ensuring the quality of fruit is of extreme importance in China as fruits have been becoming the major constituent of the Chinese diet due to their high nutritional value. The consumption of fresh fruits in China has increased during the last two decades due to the rising recognition of the nutritional value of these agricultural products. According to the China Statistical Yearbook (1991–2015), the Chinese consumed an estimated 56.1 kg of fresh fruits per urban household in 2012, up from 41.1 kg in 1990 (Figure 2), whereas 100.1 kg per urban household of fresh vegetables were consumed by Chinese consumers in 2014, down from a high of 138.7 kg in 1990. In particular, according to the National Bureau of Statistics food availability data, urban Chinese on average consumed more fresh fruit (48.1 kg per household) than meat, eggs, and nuts (41.9 kg in total) in 2014. 

Fresh apples can be used as an illustrative case to interpret consumers’ WTP a premium for organic fruits for the following reasons. China is one of the largest fresh apple producers and exporters in the world [43]. Apples are an important source of different vitamins and rich in nutrients [44]. According to the Chinese Statistical Yearbook (2007–2018), the consumption of fresh apples has increased from 25.12 million tons in 2006 to 43.50 million tons in 2017. The growth rate of fresh apple consumption has reached 70% in the past 11 years in China.

To gain a general picture of urban consumers’ willingness to pay a premium for organic fruits across major regions of China, the research was conducted in nine cities distributed in different parts of China. The nine cities partially represent the diverse types of cities in China (Figure 3): Beijing is the political and cultural capital city of China; Nanjing, Ningbo, and Wuhan, located in southern China, are characterized as developed cities in China; Zhengzhou is an average city in central China; Taiyuan and Xi’an are characterized as developing cities in northern and northwestern China; Lanzhou and Guiyang are regarded as undeveloped cities located in western and southwestern China, respectively.

Overall, from the perspective of economic development, the nine cities could represent the developed, developing and undeveloped regions in China; from a geographical view, they distributed in central, northern, southern, western parts of China. 

A self-completion paper questionnaire was used to obtain the data, coupled with a face-to-face interview. In each city, five chain supermarkets located in different parts of the city were selected, the survey was conducted from 9 to 11 am, and 3 to 5 pm in the weekend so that the collected consumers could be more representative. To guarantee the respondents were all fruit consumers, the interviews standing in front of the supermarket were required to question consumer who came from the supermarket with a receipt showing that he/she bought a kind of fruit from the supermarket. Finally, 50 urban consumers participated in the survey and a total of 450 questionnaires were collected during May–August 2014. The 450 original survey answers were entered into computer files for further analysis, and 407 respondents were eligible for the statistical analysis after the data cleaning.

A series of questions about respondents’ attitudes are questioned. All interviewees were asked a number of close-ended questions divided into three parts. The first part mainly concerned respondent’s socio-demographic characteristics including age, educational level, gender, occupation, family size, monthly income, working conditions, and location of residence, etc. The second section was dedicated to acquiring information on consumers’ preferences to fruit-specific attributes covering sales price, taste, appearance, color, variety, wrapping, province of origin, and nutritional value applying a five-point Likert scale. These statements were designed to test consumers’ attitudes toward the importance of fresh fruit attributes in their payment decisions. In the final section, respondents’ perception and knowledge of organic label, their judgment on safety food, and trust in current monitoring the situation of food quality safety issues by the State Council, ministries, and local governments at all level were investigated. Particularly, in order to figure how much a positive amount consumer willing to pay for organic fruit, the question “if you are willing to purchase organic fruits, how much more you are willing to pay for organic apples compared to non-organic apples with similar size and variety?” is indicated in the questionnaire.

### 4.2. Description of Variables

Table 1 contains a description of the variables used in the study, along with their associated summary statistics. 

More than half of the respondents (54%) were female, which was confirmed by previous literature [45,46]. This is reasonable because females are the primary shoppers for household products in China. Of the 407 respondents, 61.4% expressed WTP price premium for organic fruits, with an average premium of 0.69 yuan/kg. Moreover, the average purchase price of organic fresh apples was reported as 7.58 yuan/KGg, thus the price premium consumers willing to pay was approximately equal to 10% of the average purchase price. The average age of fruit consumer was around 39, with 16 years of formal education obtaining bachelor degree. Typically, 78% of participants had a well-educated job, with monthly income between 3000 and 5000 yuan on average. In the questionnaire, consumers were asked to rate the degree they noticed the organic label when purchasing fruits. Although most of the respondents showed highly credence in organic label, they often do not pay much attention to the label when buying fruit. Therefore, noticing an organic label only scores 2.30, but belief in the same label is 4.39.

### 4.3. Attitudes toward Fruit Safety

To assess consumer’s attitude towards fruit safety issues, a five-digit Likert scale was used, where 1 = very low and 5 = very high. Give the past research in this area [47,48,49,50], the following statements as explanatory variables in the matter of consumer’s WTP decision making are included. Figure 4 depicts the differences of consumers’ attitudes toward fruit safety between WTP and unwilling to pay (UWTP). 

In terms of fruit safety, less than 10% of consumers in both groups expressed great worry about fruit safety. 41.4% of UWTP respondents and 31.2% of WTP ones declared high attention to fruit safety issues, respectively. Concerning the notice of organic label when purchasing fruits, 16.0% of WTP consumers paid attention to the organic label compared to 9.6% of UWTP consumers, more than half of the interviewees in both group expressed no concern. 75.8% of UWTP respondents told us that they had a lower frequency of purchasing unsafely fruits, compared with 45.2% of WTP interviewees. The result suggests a positive relation between consumers’ WTP and their frequency of buying unsafely fruits. In addition, nearly 80.0% of WTP consumers had confidence in organic label whereas only 54.8% of UWTP ones expressed the same attitudes. This data reflects that the basic knowledge of organic label and the trust in organic label may significantly affect consumers’ WTP a premium for organic fresh fruits.

### 4.4. Perception of Importance of Fruit Attributes

To obtain the information on the perception of importance on fruit attributes that affect consumers WTP, an ordinal scale ranging from 1 = not important at all to 5 = very important was used. Consumer evaluations of the importance of fruit attributes were quite different between the two groups (Table 2). Taste and appearance were regarded as the most important attributes in both groups. Purchase convenience and the variety of fruits are perceived as the second most important attributes by WTP consumers (60.4%). The respondents in the WTP group rated the nutritional value (47.6%) comparatively highly while 18.5% of UWTP consumers perceived nutrition as important. On the other hand, substantial consumers in the UWTP category evaluated sales price as “very important” (72.0%). This might be partly the reason for their unwillingness to pay a higher price for organic fruits. In addition, wrapping appears to be less acute for all interviewees.

## 5. Empirical Analysis

The determinants of WTP identified through the double-hurdle model are reported in Table 3. The first two columns illustrate the effects of the fruit-specific attributes and socio-economic characteristics on the probability that a consumer express willingness to pay a premium for organic fresh fruits, while the determinants of the amount paid for organic fruits are presented in the last two columns. Marginal effects evaluated at the means for each of the explanatory variables are contained in Table 3.

### 5.1. WTP a Premium Choice Strategy

A cursory look the results displayed in Table 3 emphasizes the importance of consumer’s education, income, occupation, location of residence, purchase price, and fruits consumption frequency in explaining the probability of paying a premium for organic fresh fruits by Chinese consumers. All the above mentioned statistically significant variables have positive contributions to consumers’ WTP. 

Respondent’s educational background plays an important role in WTP, while the magnitude is small (3.69%). Better educated consumers are more likely to buy organic fruits at a higher price. Our findings have nonetheless corresponded with those of Chelang’a et al. [46] and Kapoor and Kumar [51], as the literature shows that high-income respondents have greater probabilities to be willing to purchase organic foods [45]. Likewise, the estimated results show that consumers living in developed provinces with decent jobs and better salary are more likely to pay a premium for organic fruits. The result is in line with the findings from Nandi et al. [52]. Besides, the WTP likelihood is higher for participants who frequently buy expensive fruits. A plausible explanation is that consumers expending higher on fruits may pay more attention to the quality and have a better income to afford the price. This finding is reinforced by the fact that the coefficient on income and occupation being positive and statistically significant at the 1% level. 

Consumers attaching more importance to fruit appearance, nutritional value, and organic label will show a higher WTP a premium for organic fresh fruits. The coefficients are positive and statistically significant at the 1% level. This is in consist with the findings from Tobia et al. and Wang and Huo that consumers who are more concerned about nutrition and certificate have higher willingness to pay a premium for organic food products [53,54]. Surprisingly, sales price showed a statistically insignificant impact on consumers’ WTP, probably because their incomes have reached the level to allow them to concentrate on quality and safety instead of cheap price. Similarly, wrapping, province of origin, variety and purchase convenience are unimportant attributes to consumers’ WTP. During the survey we noticed that more than 80% of the participants would value wrapping as an important attribute only when the fruit was purchased as a visiting gift. These results are crucial for fruit growers and agro-firms to conduct cost and benefit analysis during fruit planting process or before starting any consumer promotion programs for the certified policy. The significantly positive sign for the ATTE (the concern to food safety) variable indicates that the probability of consumer WTP a premium for organic fresh fruits increases as concerns about safety issues on fresh produce increase. Consumers who frequently purchase unsafety food show a higher probability to pay a premium. This result makes sense that failure purchase experiences might stimulate higher passion of consumers on buying safety food with a premium. Wu et al. found similar results [55].

### 5.2. WTP a Premium Choice and Its Impact on the Premium

Table 3 presents the parameter estimates and elasticity for the determinants affecting premium paid for organic fresh fruits. The elasticity reflects changes in the explanatory variables on the level of the amount paid by those consumers who are willing to pay a premium. As stated earlier, variables referring to consumer’s education background, income, and location of residence play a positive and significant role on the amount paying for organic fresh fruit. In other words, respondents with more schooling years, higher salary, and live in a developed province in China observe an increased payment in the premium for organic produce of 0.03%, 0.18%, and 0.11%, respectively. Misra and Ott pointed out that consumers with more formal education probably had a better understanding of the importance of organic label on fresh produce [56]. Govindasamy and Italia discovered that a better educated consumer had a higher degree of confidence in the quality certificate [57]. According to the survey data, people living in developed provinces, i.e., Beijing, Nanjing, Wuhan, were willing to pay 0.69 yuan/kg more for organic fresh fruits than those living in less developed regions such as Lanzhou and Guiyang.

Other fruit attributes that have a positive impact on the premium paid by consumers who have shown a WTP are the appearance and the organic label of fruits. Hamzaoui-Essoussi and Zahaf recognized that organic label for fruit was not only perceived as a guarantee for fruit quality, but also as an assurance of the fruit safety [58]. That is, consumers who show a higher degree of confidence in the label have a significantly greater probability of WTP a premium for organic fruit and are prepared to pay more than those taking organic label unimportant. Moreover, although the impact of other fruit attributes is positive, the contribution is small and the statistical result is insignificant.

The effective amount paid for organic fresh fruit is also significantly influenced by participants’ attention to fruit safety issues. Consumers who concentrated more on food safety issues were willing to pay 0.30 yuan/kg more for organic fresh fruits than those paid less attention to food safety. Possible implications suggest that the frequent outbreaks of serious food safety incidence indeed arouse consumers WTP a premium for fresh fruits with organic labels. 

## 6. Discussion

Previous consumer survey works based on WTP analysis gave inconsistent results on the impact of consumer characteristics on food safety assuming that consumers’ WTP is equal to their actual payment behavior. Using the field survey data from nine cities in China, this paper assesses consumer’s willingness to pay a premium for organic fresh fruits and investigates their paying behavior for the organic fruits, which are important for fresh producers and quality certificate institutions. 

Generally, it is believed that Chinese consumers have a complex attitude towards organic product largely depending on their educational background, residential location, monthly income, and attention to fruit safety. Occupation and consumption frequency significantly affects consumers’ WTP premium for organic fruits, while the influence on the amount paid by consumers for organic fruits is statistically insignificant. Income is not only significant in the participation stage, but also significant in the consumption stage. Thus, economic reasons are part of the cause of the participation and consumption of organic fresh fruits whereas social position and consumption habits are not the reasons for consumption. 

From the findings, we can construct a profile of the consumer most probably to purchase organic fresh fruits at a premium price. Better educated individuals with higher income and living in developed cities would be more likely to exhibit a greater WTP a premium for organic fresh fruits. As an implication for organic fresh fruit dealers, these findings indicate that there exist target consumer segments for the organic fresh fruit sector in China, particularly in the developed cities. That is, marketing strategies targeted at higher income, and better-educated consumers can be effective in attracting new consumers and eliciting more purchases for organic fresh fruits.

A notable result is that the degree of perception on fruit safety and the frequency of purchasing unsafely fruit are a far more compelling reason motivating a positive willingness to pay a premium for organic fresh fruit than confidence in quality certificate. It appears that the unfavorable purchase experience would be more useful to inspire consumers’ payment behavior on organic fresh fruits at a premium price than simply have trust in organic label. 

A further marketing implication drawn from the paper pertains to the importance of certification regarding the safety guarantee system through promulgating regulations and optimizing the management system. The asymmetric nature of information leads to market distortion of food quality safety problems. The recorded consumer interest in safety and quality of fresh fruits reveals that a promising market for organic fresh fruits could be developed by an adequate knowledge on organic label and an effective market supervision system. Therefore, the provision of educational programs to help consumers correctly and clearly understating various kinds of food quality certificates by both public administrations and private sectors might be an effective strategy to up consumer WTP and to rebuild his/her confidence in fresh produce certified by a regulating institution.

The findings have some limitations in terms of being able to generalize. Further research is necessary so as to enlarge the samples size to make the results be more preventative and to go ahead with design improvements to mitigate social desirability bias. Social desirability bias might lead to a greater possibility of indicating a preference for paying a higher price premium for organic apples compared to respondents’ true action. Consumers might overstate the premium to provide a socially-desirable answer. The study tried to pay more attention to identify the impact factors on consumers’ WTP premium for organic fruits, the social desirability bias is thus to be acceptable in this study. But a better method, such as non-hypothetical experiments [59] should be applied in future research. Moreover, the double-hurdle model limited in that it assumes that the shocks to the participation and consumption processes are independent, which is not always a realistic assumption [60]. The model would be improved in the future study.

## Figures and Tables

**Figure 1 ijerph-16-00126-f001:**
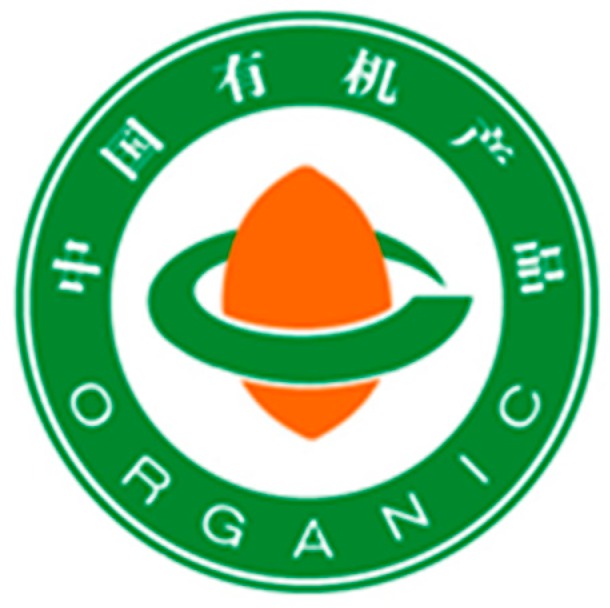
China Organic Product Certification Mark. Source: https://www.organic-bio.com/en/labels.

**Figure 2 ijerph-16-00126-f002:**
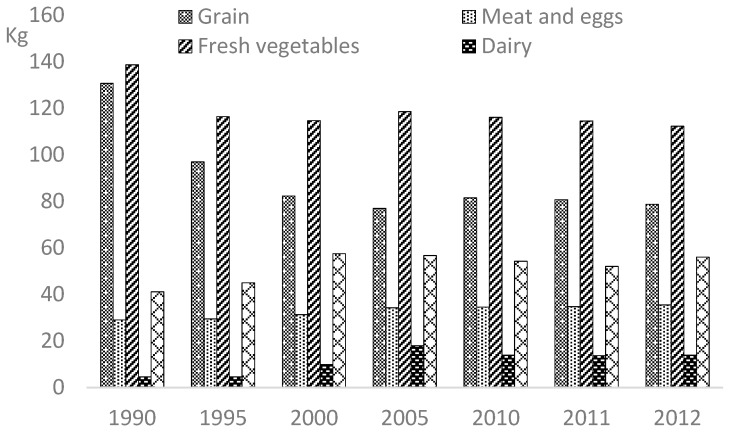
Per Capita Consumption of Major Food of Urban Households, 1990–2012. Note: The data are compiled on the basis of the integrated household income and expenditure survey of the National Bureau of Statistics of the People’s Republic of China, including urban and rural households.

**Figure 3 ijerph-16-00126-f003:**
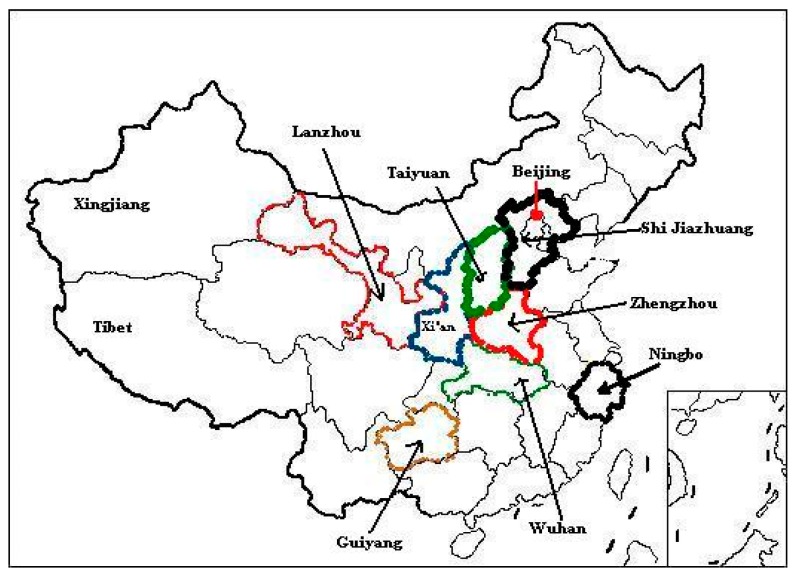
Location of the sample cities in China.

**Figure 4 ijerph-16-00126-f004:**
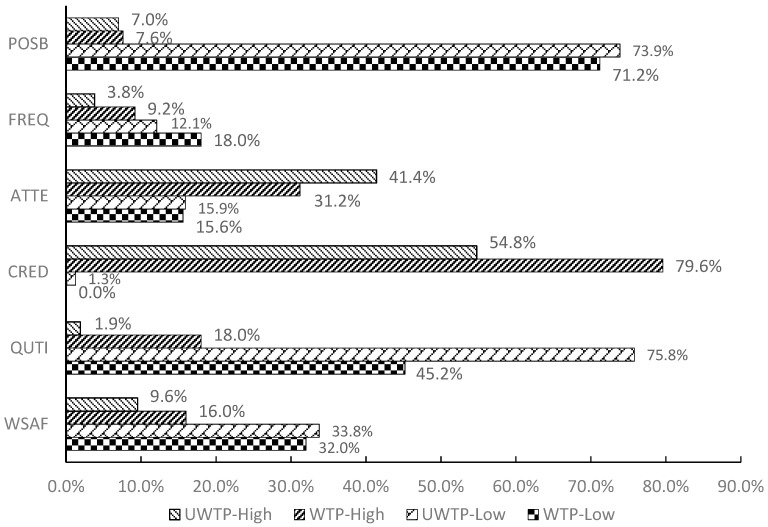
Comparison of attitudes toward fruit safety between WTP and UWTP consumers. Note: For the purpose of presentation, the five digit-scale categories were combined into three: “high” in the figure combines the frequencies of “very high” and “high” responses; “low” combines the frequencies of “very low” and “low” responses.

**Table 1 ijerph-16-00126-t001:** Summary statistics for empirical variables.

Variable	Description	Variable Scale	Mean	Std. Dev.
WTP	Willingness to pay a premium	1 = Willing to pay; 0 = Unwilling to pay	0.61	0.49
MONE	The premium pay for organic fresh fruit	yuan/kg	0.69	0.85
**Socio-economics characteristics**				
SEX	Gender	1 = male; 0 = female	0.46	0.50
AGE	Age	Years	38.96	14.66
EDU	Education ^a^	Years of schooling	16.04	2.97
SIZE	Household size	Number of individuals	3.07	0.93
INCM	Monthly income	1–5 ^b^	2.37	1.52
OCCP	Occupation	1 = Well educated; 0 = Blue collar	0.78	0.41
DEVE	Location of residence	1 = developed city; 0 = otherwise	0.49	0.50
PPRI	Purchase price	yuan/kg	7.58	3.93
CFRE	Consumption frequency	1–5 ^c^	3.57	0.97
**Importance attach to fruit attributes**				
PRIC	Sales price	1–5 ^d^	3.51	0.89
WRAP	Wrapping		2.30	0.87
APPE	Appearance		4.00	0.76
NUTR	Nutritional value		3.10	1.08
EABU	Purchase convenience		3.41	0.84
PREG	Province of origin		2.46	1.23
ORLB	Organic label		2.41	1.01
VATY	Variety		3.60	0.93
TAST	Taste		4.35	0.83
**Attitudes toward fruit safety**				
WSAF	Worried fruit safety	1–5 ^c^	1.81	0.65
QUTI	Notice of organic label		2.30	1.01
ATTE	Concerns about fruit safety		3.23	0.80
CRED	Credence in organic label		4.39	0.94
FREQ	Frequency of buying unsafely fruit which causing diarrhea or other diseases		2.83	0.94
POSB	Possibility of buying fruits produced by companies occurred safety incident		1.70	1.23

Note: ^a^ 6 = primary school, 9 = middle school, 12 = high school, 16 = bachelor degree, 19 = master degree, 22 = PhD; ^b^ 1 = less than 3000 yuan, 2 = (3000, 4000) yuan, 3 = (4000, 5000) yuan, 4 = (5000, 6000) yuan, 5 = equal and more than 6000 yuan; ^c^ 1 = very low, 2 = low, 3 = moderate, 4 = high, 5 = very high; ^d^ 1 = not importance at all, 2 = not important, 3 = moderate, 4 = important, 5 = very important.

**Table 2 ijerph-16-00126-t002:** Perception of Importance on fruit attributes. (Percent of respondents in each category) ^a.^

Item	Willing to Pay (N = 250)			Unwilling to Pay (N = 157)		
	Unimportant	Moderate	Important	Unimportant	Moderate	Important
Taste	3.2	11.6	85.2	4.5	7.0	88.5
Appearance	3.6	12.8	83.6	3.2	17.2	79.6
Purchase convenience	8.8	30.8	60.4	18.5	45.9	35.7
Variety	10.4	29.2	60.4	15.3	25.5	59.2
Nutritional Value	19.6	32.8	47.6	39.5	42.0	18.5
Sales price	16.8	40.0	43.2	13.4	14.6	72.0
Province of origin	49.2	22.0	28.8	62.4	28.7	8.9
Organic label	42.0	34.0	24.0	75.8	23.6	0.6
Wrapping	49.2	44.4	6.4	63.7	33.1	3.2

Note: ^a^ For the purpose of presentation, the five digit-scale categories were combined into three: “important” in this table combines the frequencies of “very important” and “important” responses; “unimportant” combines the frequencies of “not important at all” and “not important” responses.

**Table 3 ijerph-16-00126-t003:** Estimation results of the double-hurdle model.

Independent Variables	First Hurdle Probability of Paying		Second Hurdle Amount Paid	
	Coef. (Std. Err.)	Marginal Effect	Coef. (Std. Err.)	Elasticity ^a^
**Socio-economic characteristics**				
SEX	−0.1355 (0.2026)	−0.0492	0.0939 (0.1964)	0.0838
AGE	−0.0034 (0.0097)	−0.0012	−0.0769 (0.0498)	−0.0687
EDU	0.1018 (0.0510) **	0.0369	0.0339 (0.0107) ***	0.0271
SIZE	−0.1286 (0.1049)	−0.0467	−0.0521 (0.1086)	−0.0465
INCM	0.3769 (0.1249) ***	0.1368	0.1990 (0.0680) ***	0.1777
OCCP	0.6892 (0.2409) ***	0.2622	0.2168 (0.3441)	−0.1136
DEVE	0.6381 (0.2070) ***	0.2292	0.6895 (0.2129) ***	0.6156
PPRI	0.1633 (0.0265) ***	0.0593	0.1258 (0.0270) ***	0.1123
CFRE	0.2471 (0.0843) ***	0.0897	0.0887 (0.0896)	0.0792
**Importance attach to fruit attributes**				
PRIC	0.0323 (0.1242)	0.0117	0.0899 (0.2868)	0.0803
WRAP	−0.0979 (0.0711)	−0.0355	0.0930 (0.1166)	0.0830
APPE	0.3205 (0.1130) ***	0.1163	0.3702 (0.1245) ***	0.3305
NUTR	0.3255 (0.1020) ***	0.1181	0.0100 (0.1086)	0.0089
EABU	0.0398 (0.1004)	0.0144	0.3458 (0.1241)	0.3088
PREG	−0.0255 (0.0771)	−0.0093	0.0917 (0.1064)	0.0819
ORLE	0.5193 (0.1272) ***	0.1884	0.2698 (0.0801) ***	0.2409
VATY	0.0544 (0.0978)	0.0198	0.0959 (0.1011)	0.0856
TAST	0.2071 (0.1146) *	0.0752	0.1635 (0.1287)	0.1460
**Attitudes toward fruit safety**				
WSAF	0.0442 (0.1429)	0.0160	−0.0189 (0.1422)	−0.0169
QUTI	0.0457 (0.0994)	0.0166	−0.0289 (0.0967)	−0.0258
ATTE	0.1536 (0.1208) ***	0.0557	0.3014 (0.0999) ***	0.2691
CRED	0.3678 (0.0951)	0.1334	0.3946 (0.1238)	0.3523
FREQ	0.2429 (0.1249) *	0.0881	0.0845 (0.1824)	0.0754
POSB	−0.0470 (0.0746)	−0.0170	−0.1596 (0.3433)	−0.1425
Observations (n)		407		250
Wald $\chi^{2}$		162.12		72.82
Log pseudo-likelihood		−168.18		−214.01
Sigma ^b^				0.8954

Note: Standard errors which are robust are reported in parentheses. Marginal effects of changing the explanatory variables are evaluated at the mean of the explanatory variables; ^a^ The elasticity is calculated at the sample mean; ^b^ Sigma is the error variance; * significant at the 10% level; ** significant at the 5% level; *** significant at the1 percent level.

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
