# Peer review of "Consumer’s Willingness to Pay a Premium for Organic Fruits in China: A Double-Hurdle Analysis"

_ijerph, 2019, doi:10.3390/ijerph16010126_

Round 1
Reviewer 1 Report
Dear Authors,
Your paper about organic food in China is interesting since there are not many such studies. I think the paper would be improved if you rearrange the text and look into my comments.

Author Response
Dear Editor and Reviewer,
We really appreciate for your useful comments and suggestions on the structure and detail of the manuscript.
We have modified the manuscript accordingly, and detailed corrections are listed below:
Introduction
I like the first part of this section, the motivation. However, I think the section is too long. Besides, I do not understand why the authors discuss the WTP literature here. That should be places in the literature review section. In addition, a presentation of the organic market in China would be appreciated.
Answer: add the following presentation of Chinese organic market in the introduction part.
“With the passage of the National Organic Program (NOP) in 2002 creating uniform USDA standards, the organic food industry has increased tremendously in terms of size and product scope (Ellison et al., 2016). According to Research Institute of Organic Agriculture (FiBL), by 2014 the land area of organic agriculture in China had reached 1.9 million hectares. This is just behind Australia (22.7 million hectares), Argentina (3.1 million hectares) and the United States of America (2.0 million hectares). In 2016, the area of organic culture reached to 2.3 million hectares, compared to 1.6 million hectares in 2015. Therefore, China occupies the third place in the world regarding the area of organic agriculture (Willer and Lernoud, 2018).However, the sales volume of organic products in domestic China has been continuously increased (Source: Certification and Accreditation Administration of the People’s Republic of China, CNCA). By the end of 2015, the sales volume of organic products in domestic China had reached 129.9 billion RMB, 10.95 thousand production enterprises received certification which was officially approved by National Standard GB/T19630 Organic Products (CNCA, 2017).”
Econometric model
It is good that the benefit of the model is listed, but I would like to know the drawbacks as well. Besides, I think the equations could be dropped. They are quite standard, but more importantly, they are not used later in the text.
Answer: the author revised the Econometric model part. We think it might be better to illustrate the basic equations of the models although they are standard. Thank you for the suggestions.
Design of study
1. Please describe why the different cities are selected. Do you expect to find different results between the cities?
Answer: Overall, from the perspective of economic development, the nine cities could represent the developed, developing and undeveloped regions in China; from a geographical view, they distributed in central, northern, southern, western parts of China. Actually, we do expect to find different results among these cities, but this would be in another manuscript.
2. The design of study also includes results (in table 1). I find that a bit strange. I do not expect to find results in a section called “Design of study”. More importantly is that the results are not commented. Some results are strange to me. How could it be that awareness of organic label only scores 2.30, but the credence in the same label is 4.39? I think it is very strange that the consumers have credence in something they hardly know.
Answer: The design of study is separated into two parts; the design of study includes data collection and questionnaire design. The descriptive results are put in part 4.
In the questionnaire, the respondents were asked to rate the degree of noticing the organic label when purchasing fruits. Although most of the respondents showed highly credence in organic label, sometimes they do not pay much attention to the label when buying fruit. Therefore, awareness of organic label only scores 2.30, but the credence in the same label is 4.39.
Empirical analysis
1. Good presentation of the econometric results, but the table is probably too big in order to get a proper overview. It is difficult to see what is important.
Answer: sorry for the big table. It was supposed to be two tables, but as the variables are the same in two tables, the two stage results were combined together in one table.
2. Discussion
It is good that the paper summarizes the method and the findings from the study. However, I would appreciate some sentences about the problematic issues in the paper, especially Chinese consumers’ knowledge of organic food. You ask questions about such food, but the knowledge seems to be very low. I guess that make the questions quite hypothetical for the respondents, and I would like a discussion of hypothetical bias. In addition, you only ask questions, so the answers do not influence the respondents in any way. Here I suggest the authors look into the literature of social desirability bias, see Fisher (1993) and Norwood & Lusk (2011).
Answer: Thank you for reviewer promoting this question. This is a limitation of our experimental design. We did not consider the problem of social desirability bias during the questionnaire design process. We suppose the respondents are rational to express their attitudes accurately, and their behaviors on paying premium to organic fruits would not be affected by the social pressure and psychological embarrassment. In the future study, we would be necessary to improve the experimental design.
Minor issues
P. 1: Instead of «panic», it should be «panicked».
Edited.
P. 1: It is said that “such as organic food, which is based on a credit mechanism”. What is meat with that?
The first two paragraphs were revised.
P. 2: Very good that the Chinese organic label is shown. However, I need to know much more about this label. I would like to know a short history of it, how many products are labelled (i.e. market share), who issues the label etc. Above all I would like to know how this label stands in comparison with the two leading organic labels in the world, the USDA Organic and the EU “leaf label”.
Answer: the introduction section was revised, please check the revised manuscript.
P. 2: The sentence “Unlike genetic modified food with much argument, organic food is more acceptable by consumers in the first place” needs to have a reference and it should be written in which country the study is conducted.
The first two paragraphs were revised.
P. 2: When reviewing the literature, I would like to see more Chinese studies because they are more relevant in a Chinese setting. For example, I think Ayyub et al. (2018) and Zheng et al. (2013) could be worth mentioning.
A literature review section was added in the manuscript. The references were mentioned in the text, and other literatures about China are added as well.
P. 6: You state that you try to make the sample as representative as possible by using several supermarkets, different time of day etc. That is good, but I would like you to extend table 1 to also describe the sample compared with the Chinese population, at the best in the selected cities. If this is present the reader can judge how representative the sample actually is. Besides a foreign reader will have trouble interpreting the numbers in table 1, are these numbers low or high?
The urban population of China at the end of 2014 was 0.75 billion persons according to China Statistical Yearbook 2017. The sample size is very small compare with the Chinese population. It seems to be unrealistic to cover most of the people living in the cities, because we use face-to-face interview coupled with questionnaire. We would enlarge the samples size in the future study with internet survey which would be better to enlarge the sample size.
P. 6: Your N seems to be 407. That is small. Please comment on why the sample is small.
We plan to cover more cities in different part of China, and meanwhile with various economic development levels. But Due to the time and fund limitation, finally the sample size is small. This is also the limitation of the manuscript. We would definitely enlarge the sample size to make the results be more representative in the future study.
P. 6: In the introduction you mention food scandals. This is one of the motivations for Chinese consumers to buy organic. I totally agree upon that, but could not also the scandals could also motivate increased consumption of foreign fruit? My question is then, why do you not have a variable for domestic vs. foreign fruit?
The sample regions cover developed, developing and undeveloped cities, the consumption of foreign fruits of consumers living in developing and undeveloped cities is small. Moreover, we do design questions related to foreign fruits, but in the pre-research phrase, almost all the respondents said that the foreign fruits would not affect their purchasing behavior on organic fruits due to the extremely high price. Further, China is one of the largest fresh apple producer and exporter in the world. In the market, the foreign apples are very rare.
P. 7: I do understand that females buy the most household products in China, I guess that the same all over the world. However, I do not understand why that is a reason to oversample the females. I guess, both genders eat organic fruits, so the men’s share in the sample should be equal to the females’ share.
The respondents were randomly selected. In the stage of data analysis, we found that female respondents account for slightly more than male respondents. The result is similar with other studies in different countries.
P. 7, table 1: What is the difference between “Concerns about fruit safety” and “Worried fruit safety”?
“Concerns about fruit safety” means “paying much attention to fruit safety and fruit safety issues”; “Worried fruit safety” means “being more anxious on fruit safety issue and afraid of buying unsafe fruit”
P. 12: I think it is a bit exaggerating to state you have “national wide data”. You only have 407 respondents and these respondents are only from nine cities. The rural population is for example not present in the sample.
The authors totally accept the reviewer’s comment. The comparatively small sample size is the big limitation of the paper. We revised the sentence as “Using the field survey data from 9 cities in China,”
P. 12: What is meant by “participation stage in the sentence “Occupation and consumption frequency are only significant in the participation stage for fresh fruits”?
In table 3, variables of “occupation” and “consumption frequency” significantly affect consumers’ WTP premium for organic fruits while the influence on the amount paid by consumers for organic fruits is statistically insignificant.

Reviewer 2 Report
See PDF

Author Response
We really appreciate for your useful comments and suggestions on the structure and detail of the manuscript. We have modified the manuscript accordingly, and detailed corrections are listed below: Introduction 1. The key article point is whether and in what magnitude Chinese consumers have a willingness to pay a premium for organics. Once the article captures that, the willingness to pay could be compared to the additional costs of organic production in order to allow a cost benefit analysis. In addition, as Chinese Government has set up policies toward organics, the costs of these policies could be compared to the potential welfare effects of organics. Therefore, a discussion on benefit-cost analysis should be in the introduction; Answer: the aim of the paper is to investigate consumers’ WTP premium for organic fruits. In the introduction, we focus on the description of the food safety issues and the development of organic food in China. We will leave the discussion of benefit-cost analysis in the future study. Thank you for the suggestions. 2. Organic food and food security in China should be better connected; Answer: the introduction is re-written to better connect the organic food and food safety in China. Please check the manuscript. 3. There are more modern methodology options in the literature (GALLARDO et al., 2017; IKIZ et al., 2017; ZHENG et al., 2016). The authors should better discuss the weakness and strengths of their methodology option. Answer: the authors add a literature review section, please check the revised manuscript. 4. There is no need of figure 1 in the text. Furthermore, it is not well placed within the text. Answer: the authors thought that the organic mark shown in the manuscript might help the readers to get the image of Chinese organic label. It was re-placed within the text. Econometric Model The normal density function Greek letter appears to be in capital form to me. This implies a accumulated function instead. Furthermore, the equations are messy around the text. Answer: the equations are re-edited. Please check the revised manuscript. Empirical Analysis Authors present marginal effects in this section. It is not clear at the methodological section how the marginal effects were computed. Answer: Because the other reviewer pointed that the equation are quite standard, might be dropped. So in terms of the marginal effects, an explanation is added in the Econometric section, and the references are added. If some readers want to know the detail of the marginal effects, they could look up the references. The added contents: “In order to interpret as the percentage changes in probability that consumer is willing to pay a premium for organic fresh fruits due to a unit change of the explanatory variable of interest given others being constants, marginal effects are calculated in the binary Probit model [Anderson and Newell, 2003]. With regard to dummy explanatory variables, the marginal effect evaluates the probability changes caused by a discrete change from 0 to 1 [Greene, 2003]”. Author Contributions The contribution of X.H. is not shown. Answer: revised: “J.W. and X.H. revised the manuscript”.Reviewer 3 Report
This study applied a double-hurdle model to analyze Chinese consumers' WTP for organic fruits. I have a few comments questions for the authors.
The authors need to clearly identify their contributions to the literature. To do so, they first need to clearly identify their objectives. For example, does the paper try to develop marketing strategies for organic producers, or to associate organic food with food safety? Is the application in a specific food category (fruit) the major contribution of this study?
The research design needs to be explained more clearly. For example, how was the purchase price for organic apples reported? was there a question in the survey? was the price from their purchase receipt? Was organic apple the only fruit asked in the survey? If respondents report organic apple price because they purchased before, how many of them had previous purchasing organic produce experience. The previous experience could be really important for their decision making, and if the authors have such information, they might want to include it as a variable. If it's possible, including sample questions might be helpful.
Do the authors ask respondents why or why not they are willing to purchase organic produce? How did you measure their knowledge about organic products? Did respondents associate organic with food safety?
How do you define "frequency of buying unsafety (*unsafe)" fruits? I don't quite understand this phrase.
Please talk about the limitations of the methods and design.
Author Response
We really appreciate for your useful comments and suggestions on the structure and detail of the manuscript.
We have modified the manuscript accordingly, and detailed corrections are listed below:
1. The authors need to clearly identify their contributions to the literature. To do so, they first need to clearly identify their objectives. For example, does the paper try to develop marketing strategies for organic producers or to associate organic food with food safety? Is the application in a specific food category (fruit) the major contribution of this study?
Answer: Results of the study should help organic fruits producers and retailers to adjust marketing strategies according to market demand. Understanding consumers’ perception and attitude toward organic fruits would provide insight for policymakers in formulating regulation to restore consumer confidence. Overall, the article aims to understand consumers’ WTP a premium price for organic fresh fruits taking fresh apple consumers as an example, and elicit factors affecting their payment behavior.
2. The research design needs to be explained more clearly. For example, how was the purchase price for organic apples reported? was there a question in the survey? was the price from their purchase receipt? Was organic apple the only fruit asked in the survey? If respondents report organic apple price because they purchased before, how many of them had previous purchasing organic produce experience. The previous experience could be really important for their decision making, and if the authors have such information, they might want to include it as a variable. If it's possible, including sample questions might be helpful.
Answer: we do design a question “Do you have experience on buying organic apples? If yes, how much more you spent on buying organic apples compared to non-organic apples with the similar size and variety?” in the questionnaire. But the number of the “yes” answer accounts for less than 10% of the total sample. So we abandon this variable in the hurdle model.
3. Do the authors ask respondents why or why not they are willing to purchase organic produce? How did you measure their knowledge about organic products? Did respondents associate organic with food safety?
Answer: in the questionnaire, we only asked respondent “do you willing to pay much more money for organic fruits? If “yes”, taking apple as an example, how much more will you going to pay for organic apple compared to non-organic apples with the similar size and variety?” we did not ask respondents why or why not they are willing to purchase organic produce? We would consider this part in the future study.
Before asking questions in the questionnaire, we firstly showed the organic label to the respondents to ask them whether knew the label or not. If the answer is “yes”, we assumed the respondents had knowledge about organic products. If the answer is “no”, we explained to the individuals.
According to our survey and face-to-face interview, respondents take organic fruits as safety fruits without pesticide and chemical additive.
4. How do you define "frequency of buying unsafety (*unsafe)" fruits? I don't quite understand this phrase.
Answer: in the questionnaire, the phrase “unsafely fruits” points to those which causing diarrhea or other diseases. In terms of variable “frequency of buying unsafely fruits”, we asked the respondents that how often consumers purchase those fruits causing diarrhea or other diseases.
5. Please talk about the limitations of the methods and design.
Answer: The findings bear some limitation in terms of being able to generalize. Further research is necessary so as to enlarge the samples size to make the results be more preventative and to go ahead with design improvements to mitigate social desirability bias, such as using non-hypothetical experiments [57]. Moreover, the double-hurdle model limited in that it assumes that the shocks to the participation and consumption processes are independent, which is not always a realistic assumption [58]. The model would be improved in the future study.
Round 2
Reviewer 1 Report
Dear Authors,
Thank you for providing av new version of the paper "Consumer’s Willingness to Pay a Premium for Organic Fruits in China: A Double-Hurdle Analysis" (ijerph-394702).
You have improved the paper a lot and have taken many of may comments in consideration. That is good.
What I feel is missing is a deeper discussion of the weakness with the method. It is very good you have added some sentences at the end of the paper about this, but I do not think it is sufficient. I would like to know how you imagine that these weaknesses affect the results. Referring to one of my suggested references is not sufficient, as I see it.
In p. 9 you have added the sentence "supposing the respondents are rational …". This assumption is clearly wrong, as there is a huge literature on how respondents answer in a survey like this. The most important effect is that they tend to give a higher WTP estimate compared to what they actually are willing to pay. Of course, assumptions are necessary, but they need to be as realistic as possible and justified. At the end of the paper I would like a discussion of why you add this assumption if you intend to keep it.
In addition, I think the English must be improved. The sentence structure is often quite weird. Hence, I suggest some sort of proofreading.
Based upon this, I think some revisions are necessary.
Author Response
Dear Reviewer,
Thank for your precious comments on the draft. We have modified the draft accordingly, and detailed corrections are listed below:
1. What I feel is missing is a deeper discussion of the weakness with the method. It is very good you have added some sentences at the end of the paper about this, but I do not think it is sufficient. I would like to know how you imagine that these weaknesses affect the results. Referring to one of my suggested references is not sufficient, as I see it.
Answer: add the weakness of the method in the final paragraph.
Social desirability bias might lead to a greater possibility of indicating a preference for paying a higher price premium for organic apples compared to respondents’ true action. Consumer might overstate the premium to provide a socially-desirable answer. The study tried to pay more attention to identify the impact factors on consumers’ WTP premium for organic fruits, the social desirability bias is thus to be acceptable in this study. But a better method, such as non-hypothetical experiments [57], should be applied in the future research.
2. In p. 9 you have added the sentence "supposing the respondents are rational …". This assumption is clearly wrong, as there is a huge literature on how respondents answer in a survey like this. The most important effect is that they tend to give a higher WTP estimate compared to what they actually are willing to pay. Of course, assumptions are necessary, but they need to be as realistic as possible and justified. At the end of the paper I would like a discussion of why you add this assumption if you intend to keep it.
Answer: Thank you for the reviewer to point this important issue. We decide to delete the sentence. We will pay much attention to the basic assumptions before designing questionnaire in the future study.
3. In addition, I think the English must be improved. The sentence structure is often quite weird. Hence, I suggest some sort of proofreading.
Answer: the English was improved.
Reviewer 2 Report
The current version is not well formatted and I cannot be sure where the changes were made. For instance, the previous version does not have a Literature Reviews (sic) section. Therefore, all text appearing in this section should be in red by MS Word standarts. Some of them are in black.
I suggested an English review and the authors do not mention and provide a proof the article was reviewed. The funding section is poorly written as well.
Notwithstanding these issues, the article has improved. If the authors are fast enough to address the above mentioned issues, the article should be considered for publication. But I want to highlight that these issues are mandatory for my aceptance.
Author Response
Dear Reviewer,
Thank for your precious comments on the draft. We have modified the draft accordingly, and detailed corrections are listed below:
The current version is not well formatted and I cannot be sure where the changes were made. For instance, the previous version does not have a Literature Reviews (sic) section. Therefore, all text appearing in this section should be in red by MS Word standards. Some of them are in black.
Answer: The authors apologize for the confused revision in Literature reviews. The reason of part of the text appearing in black in this section is that we use the comparative button in MS Word to show the differences between the revised draft and the original draft. Some of the sentences in Literature reviews in the revised version were also in the original draft.
I suggested an English review and the authors do not mention and provide a proof the article was reviewed. The funding section is poorly written as well.
Answer: revised the funding part.
Funding: We thank the three anonymous reviewers for their constructive comments on an earlier draft. Financial support was provided by the Key Projects of the National Natural Science Foundation of China (Grants 71673115), the Belt and Road Special Project of Lanzhou University (Grants 2018ldbrzd006), and the China Agriculture Research System (Grants CARS-28).
The English was improved.

Reviewer 3 Report
I know there are limited changes that the authors can make based on the existed survey, and I appreciate their efforts on the revision. I don't have further questions.
Author Response
The authors appreciate the reviewer's constructive comment. Thank you very much !